# Far-field phonon coupling in valley metamaterial circuits

Yao Huang[1], Weitao Yuan [2], Zhiwei Guo [3], Qi Wang [4], Yuxuan Zhang[1], Yueting Zhou [1], Yongdong Pan[1], Jia Zhou [5], Oliver B. Wright [6], Zheng Zhong[7] & Jinfeng Zhao [1] ✉

On-chip whispering-gallery-mode cavities enable versatile bosonic wave manipulation but typically rely on near-field evanescent coupling. Here we experimentally demonstrate broadband far-field phonon coupling in a valley metamaterial cavity integrated with a Dirac-cone waveguide—termed a "Dirac strip". The far-field coupling is confirmed by transmission spectroscopy and spatiotemporal field mapping over distances up to approximately five wavelengths, enabling multiplexed, distance-robust coupling pathways that overcome near-field limitations. By combining near-field and far-field cavities on the same substrate, we achieve amplification and control of non-Hermitian dynamics through loss and distance modulation of sympathetic resonances, directly resolved via piezo-laser interferometry. This work establishes a scalable phononic platform for far-field coupling and paves the way for parallel topological wave processors.

Whispering gallery modes (WGMs), characterized by continuous reflections along curved cavity interfaces[1], have become foundational elements in waveguide-cavity coupling systems with broad utility across acoustics[2], elastodynamics[3], optics[4], and electronics[5]. These modes enable key functionalities including low-threshold lasing[6], enhanced sensing[7], and critical coupling – where dissipative loss ($\Gamma$) equals radiative loss ($\gamma$)[8]. The pursuit of advanced on-chip coupling schemes has garnered considerable interest in scalable information processing[3–5,8].

Recent advances in enhancing waveguide-cavity coupling have exploited non-Hermitian physics in waveguide-cavity systems through deliberate modulation of external loss and gain. This approach enables phenomena such as exceptional points[9,10] that yield nonlinear dynamics[11], nonreciprocal transmission[12], and spectral flows[13,14]. Non-Hermitian systems are non-conservative and characterized by energy exchange[15,16], encompassing three situations: loss only, gain only, or both. Concurrently, topological metamaterials have transformed

continuum waveguide and cavity design, giving rise to topological edge states (TESs)[17,18], topological whispering gallery modes (TWGMs)[19–21], and corner states[22,23]. These bosonic wave counterparts encode topological protection and synthesized chirality[24,25], liberating cavity designs from perfect circular geometries[26,27]. Valley-phononic TWGMs can exhibit Hermitian[20] or non-Hermitian behaviour through unit cell-level modulation of gain and loss[26].

Despite these advances, achieving far-field coupling remains a fundamental challenge in waveguide-cavity systems[28]. All approaches for coupling in such systems to date rely on evanescent near-fields, which intrinsically confine energy transfer to sub-wavelength distances. Far-field coupling schemes, by contrast, if achievable would enable unprecedented spatial flexibility for on-chip integration, full utilization of wave degrees of freedom, and direct observation of inter-cavity dynamics in multi-resonator architectures. Yet their construction has not been possible owing to the absence of broadband, robust, and distance-insensitive coupling mechanisms.

[1]School of Aerospace Engineering and Applied Mechanics, Tongji University, 100 Zhangwu Road, Shanghai 200092, China. [2]Sichuan Province Key Laboratory of Advanced Structural Materials Mechanical Behavior and Service Safety, School of Mechanics and Aerospace Engineering, Southwest Jiaotong University, Chengdu, Sichuan 611756, China. [3]MOE Key Laboratory of Advanced Micro-Structured Materials, School of Physics Science and Engineering, Tongji University, Shanghai 200092, China. [4]School of Microelectronics, Nanjing University of Science and Technology, Nanjing 210094, China. [5]State Key Laboratory of ASIC and System, School of Microelectronics, Fudan University, Shanghai 200433, China. [6]Hokkaido University, Sapporo, Hokkaido 060-0808, Japan. [7]School of Science, Harbin Institute of Technology, Shenzhen 518055, China. ✉e-mail: jinfeng.zhao@tongji.edu.cn

In this work, we experimentally demonstrate broadband, robust waveguide-cavity far-field coupling. This is achieved through topological phonon interactions in on-chip valley waveguide-cavity systems using a Dirac-cone waveguide strip (a 'Dirac strip'), which enables long-distance coupling beyond conventional near-field limits and markedly improves coupling efficiency. This topological channel mediates enhanced waveguide-cavity coupling through the broadband and robust features of valley metamaterials. By co-locating near- and far-field cavities, we observe sympathetic resonances—simultaneous resonance in both cavities at a single frequency—and reveal non-Hermitian dynamics modulated by material loss and inter-cavity distance.

## Results

### Valley metamaterial circuits: three configurations

Three valley metamaterial circuits are investigated in this work (Fig. 1a). Sample 1 is a far-field cavity (FFC) waveguide circuit composed of triangular pillars, with side length $s_l = 500\,\mu m$ and height $h_p = 292\,\mu m$ (see Fig. 1b). Samples 2 and 3 correspond to the near-field cavity (NFC) and dual-cavity waveguide circuits, respectively, with $s_l = 526\,\mu m$ and $h_p = 289\,\mu m$. All samples share the same lattice constant $a = 641\,\mu m$ and were fabricated from Si(100) wafers of thickness $e_t = 525\,\mu m$, with geometrical parameters detailed in Supplementary Table 1. The rhombus-shaped cavities have side length $7a$, and the waveguides span $38a$ horizontally. Rightward terminal boundaries are oblique to suppress reflections. The Dirac strips have a horizontal width of $3a$, extending

vertically 7 layers for FFC structures and 3 layers for NFC structures. In dual-cavity configurations, the Dirac strips of NFC and FFC are positioned with a $7a$ (center-to-center) horizontal separation. Simplified models of the three samples are shown in Supplementary Figs. 2 and 3.

We fabricated valley phononic crystal (PnC) plates by etching arrays of triangular pillars in a honeycomb lattice (see Methods). Dirac cones appear at the K point of the Brillouin zone for $\theta = 0°$ (Fig. 1c, d, upper panels), although the intrinsic elastic anisotropy of silicon opens a very small gap at the Dirac point (insets). Rotating the pillars to $\theta = 20°$ (phase A) and $\theta = -20°$ (phase B) generates broadband gaps for antisymmetric plate waves (Fig. 1c, d, lower panels), with slight Dirac-cone frequency differences between Sample 1 and Samples 2 and 3.

The unit cell of triangular pillars possesses $C_{3v}$ symmetry when $\theta$ equals zero, and topological phase transitions are induced by changing the rotation angle $\theta$ to break this symmetry. These valley PnCs support the topological states reported previously[24,29]. We adopt this classic valley-phononic paradigm and fabricate these samples to explore novel mechanisms of wave propagation.

### Single-cavity far-field waveguide system

Our first on-chip waveguide-cavity phonon system integrates the FFC positioned beside the waveguide and coupled to it via a Dirac strip of length $10 \times 0.866a$, thereby enabling the observation of far-field coupling. PnCs A and B exhibit opposite topological phases (Supplementary Fig. 4), and their interface supports topological edge states (TESs,

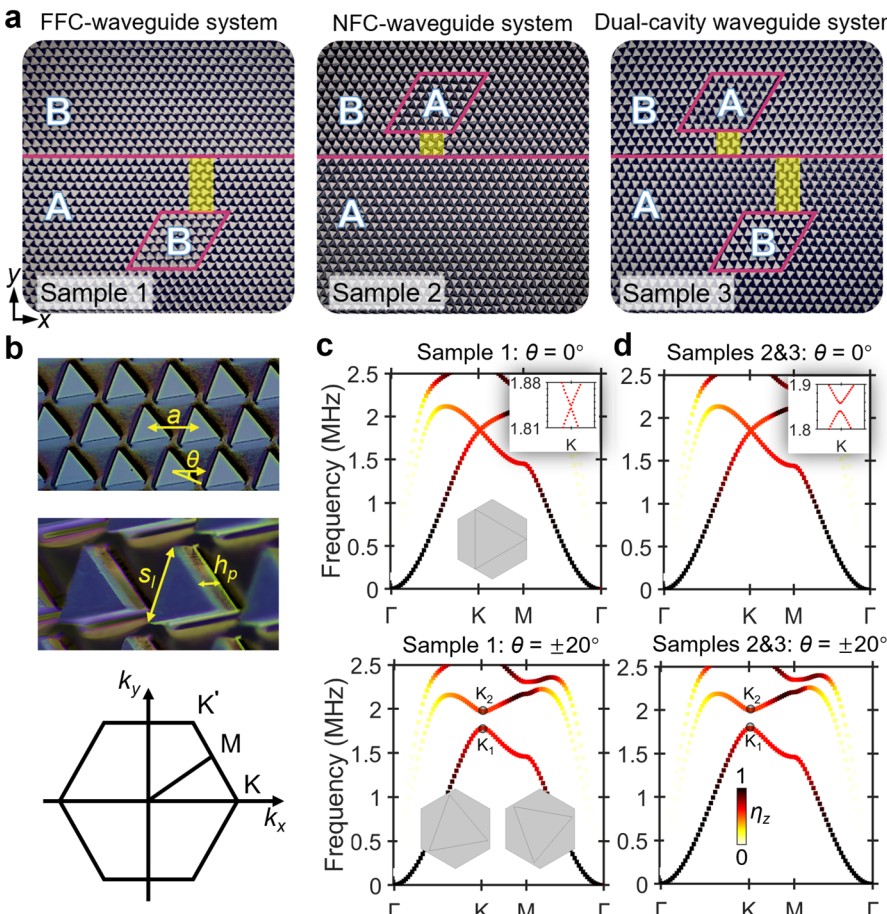

**Fig. 1 | Samples and dispersion relations for the valley phononic plates. a** Experimental samples for the (1) FFC-waveguide, (2) NFC-waveguide, and (3) dual-cavity waveguide circuits. The crystallographic directions [100] and [001] of silicon are aligned with the $x$- and $z$-axes, respectively. **b** (top) PnC plate, (middle) unit cell, and (bottom) Brillouin Zone taken from Sample 1. Geometrical parameters include lattice constant $a$, pillar side length $s_l$, pillar height $h_p$, silicon wafer thickness $e_t$, and substrate thickness $e_s = e_t - h_p$. Topological phases are controlled by tuning the rotation angles $\theta$ of the triangular pillars. **c, d** Band structures of the unit cell for $\theta = 0°$ (top panels) and $\theta = \pm 20°$ (bottom panels) are shown for Sample 1 in (**c**) and Samples 2 and 3 in (**d**). The color scale indicates the ratio of $|u_z|$ to the total displacement $|u_t|$ in the unit cell, defined by $\eta_z = \int |u_z| dV / \int |u_t| dV$. Insets show unit cell schemes and a zoom-in of the dispersion for $\theta = 0°$.

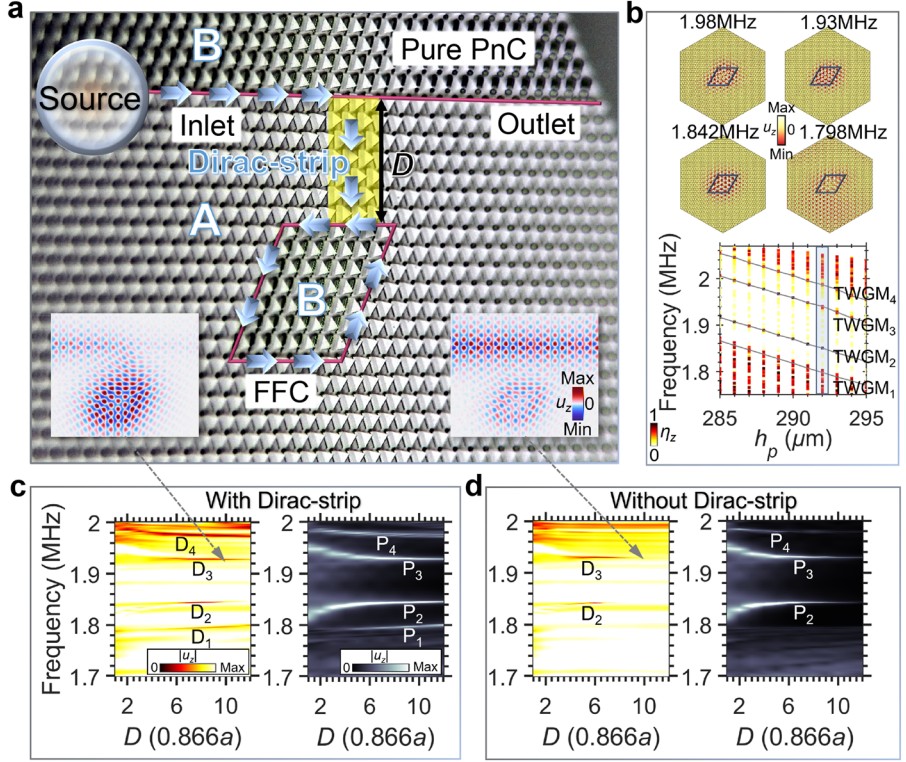

**Fig. 2 | Far-field critical coupling in a Dirac-strip-enhanced waveguide-cavity phonon system. a** Optical micrograph of the on-chip far-field cavity (FFC) connected to straight waveguide via a Dirac strip ($D = 7 \times 0.866a$, highlighted in yellow). Adjacent topologically distinct phononic crystals (PnCs) A and B define interfaces (red lines), along which blue arrows indicate the incident wave path. **b** Top: $u_z$ distributions for TWGM$_1$-TWGM$_4$ when $h_p = 292\,\mu m$. Bottom: TWGM eigenfrequencies as a function of the pillar height $h_p$ with $s_l = 500\,\mu m$. The

colour map represents the ratio of $|u_z|$ to the total displacement $|u_t|$ in the supercell. **c** Numerically computed map of normalized $|u_z|$ at the waveguide outlet (left) and cavity bottom edge (right), plotted against frequency and Dirac-strip length $D$ (i.e., cavity-waveguide distance), with the Dirac strip shown in (**a**). Left inset (shown in (**a**)): field distribution showing enhanced coupling at 1.93 MHz ($D = 10 \times 0.866a$). **d** As (**c**) but with a PnC-A domain instead of the Dirac strip. Right inset (shown in (**a**)): map of $u_z$ at 1.93 MHz for $D = 10 \times 0.866a$, in the undercoupled state.

Supplementary Fig. 5) and topological whispering gallery modes (TWGMs, Fig. 2b and Supplementary Fig. 6).

The cavity eigenfrequencies result from embedding PnC-B within a PnC-A domain (Fig. 2b). Within the antisymmetric plate wave band gap, four TWGMs (TWGM$_1$-TWGM$_4$) are identified (black solid lines), exhibiting a decrease in frequency with increasing $h_p$. For the experimental sample ($h_p = 292\,\mu m$), the TWGM$_1$ at 1.798 MHz couples to bulk modes, evidenced by $u_z$ field leakage beyond the cavity. In contrast, TWGM$_2$ (1.842 MHz), TWGM$_3$ (1.93 MHz) and TWGM$_4$ (1.98 MHz) remain confined within the band gap of both PnCs A and B, with $u_z$ distributions localized along the cavity path.

To characterize the far-field phonon coupling, we show in Fig. 2c numerically computed maps of normalized $|u_z|$ at the waveguide outlet (left panel) and cavity edge (right panel) plotted versus frequency and Dirac-strip length $D$ (i.e., the cavity–waveguide distance), obtained from finite-element simulations (see Methods), using a model matching the experimental sample in Fig. 2a. With the Dirac strip ($\theta = 0°$) bridging the cavity and waveguide, four outlet $|u_z|$ minima (D$_1$-D$_4$) emerge and shift toward the TWGM$_1$-TWGM$_4$ eigenfrequencies as $D$ increases, whereas four cavity $|u_z|$ maxima (P$_1$-P$_4$) simultaneously develop, confirming TWGM generation. These TWGMs mediate enhanced coupling at large $D$−e.g., the outlet minimum labelled D$_3$ reaches a normalized value of $|u_z| = 0.27$ at $D = 10 \times 0.866a$ and 1.93 MHz (see the left inset in Fig. 2a, corresponding to Fig. 2c). By comparison, Fig. 2d shows Dirac-strip-free counterparts−in which the Dirac strip is replaced by a PnC-A domain −where only minima D$_2$ and D$_3$ persist, with D$_1$ and D$_4$ absent. These minima vanish rapidly with increasing $D$ owing to the diminished amplitude of TWGM$_2$-TWGM$_3$, yielding poor energy transfer (e.g., at

$D = 10 \times 0.866a$ and 1.93 MHz, see the right inset in Fig. 2a, corresponding to Fig. 2d).

This enhanced far-field coupling depends on the cavity dissipation loss $\Gamma$, radiation loss $\gamma$, and resonance angular frequency $\omega_0$ (Supplementary Fig. 2). The transmission coefficient $T = |1 - \frac{\gamma}{i(\omega - \omega_0) + \Gamma + \gamma}|$ reaches a minimum value $T_{\min} = \frac{\Gamma}{\Gamma + \gamma}$. With the Dirac strip (Supplementary Fig. 7), the effect of the value of $\gamma$ dominates over that of $\Gamma$ even at large $D$, amplifying far-field coupling. This enhancement stems from the broadband, robust transport properties associated with Dirac-cone bulk states in valley-structured phononic crystals[30,31], exploited here in the Dirac strip. The modeled antisymmetric bulk-wave branches are spatially confined to the Dirac strip, and their frequency ranges overlap with the eigenfrequencies of TWGM$_1$, TWGM$_3$, and TWGM4 (Supplementary Fig. 8). These guided bulk-wave branches are spectrally tuned by Dirac-strip pillar rotation. For example, at a Dirac strip pillar angle $\theta = 10°$, these bulk-wave branches overlap only with the TWGM$_4$ eigenfrequency, directly shaping the D$_1$–D$_4$ and P$_1$–P$_4$ normalized $|u_z|$ versus $\theta$ profiles (Supplementary Fig. 8).

In situ time-resolved observations are performed by spatially mapping $u_z$ over the waveguide-cavity circuit (see Methods). Normalized $|u_z|$ is measured at representative positions: Pure PnC (i.e., unmodified PnC), FFC, Inlet, and Outlet (Fig. 2a). As shown in Fig. 3a, the FFC $|u_z|$ (blue curve) exhibits peaks P$_1$-P$_4$ at 1.808, 1.867, 1.937 and 1.976 MHz, corresponding to the calculated frequencies of TWGM$_1$-TWGM$_4$. Compared to the Pure PnC case (gray background), the outlet $|u_z|$ (red curve) shows distinct minima at the branches labelled D$_3$ and D$_4$, where the normalized $|u_z|$ reaches approximately 0.32 (1.937 MHz) and 0.30 (1.976 MHz), respectively.

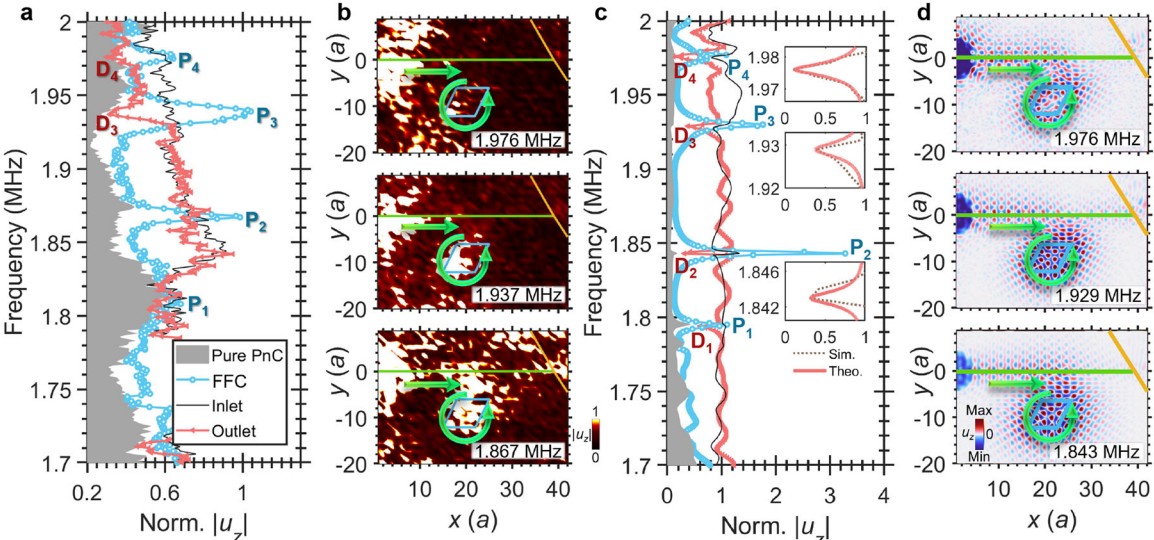

**Fig. 3 | Enhanced coupling in a far-field cavity-waveguide phonon system.**
**a** Normalized $|u_z|$ measured at locations on the sample as marked in Fig. 2a (Pure PnC, FFC, Inlet, Outlet), excited by a pulsed source centered at $f_c$ = 1.8 MHz. FFC spectrum (blue curve) shows peaks $P_1$-$P_4$ at 1.808, 1.867, 1.937 and 1.976 MHz, respectively. **b** Measured $|u_z|$ distributions at frequencies corresponding to $P_2$-$P_4$, together with the in-plane wave flux (bold arrows). Lines outline the waveguide,

cavity path, and truncated terminal. **c** Simulated spectra showing FFC peaks $P_1$-$P_4$ (blue) and outlet dips $D_1$-$D_4$ (red) at 1.794, 1.843, 1.929 and 1.976 MHz. Insets show zoomed-in views of the outlet $|u_z|$ at $D_2$-$D_4$: numerical data are shown by the dotted lines, and theoretical results by the red solid lines. **d** Simulated $u_z$ distributions at frequencies corresponding to $P_2$-$P_4$ with in-plane energy flux (bold arrows).

The $|u_z|$ field maps confirm incident wave coupling from waveguide to cavity (Fig. 3b). Minimal energy reaches the truncated outlet at frequencies corresponding to $D_3$ and $D_4$ owing to $TWGM_3$ and $TWGM_4$ excitation, whereas partial wave leakage at $D_2$ prevents outlet dip observation. The map of $|u_z|$ at $D_1$ is shown in Supplementary Fig. 9, with generation of $TWGM_1$ and bulk waves in the Pure PnC. The time-resolved $u_z$ map reveals anticlockwise circulation along the cavity path (Supplementary Movie S1).

Figure 3c, d show numerical counterparts to Fig. 3a, b. In Fig. 3c, FFC $|u_z|$ (blue curve) exhibits peaks $P_1$-$P_4$ consistent with experimental results (<0.024 MHz frequency shift). Outlet $|u_z|$ (red curve) shows dips $D_1$-$D_4$ from $TWGM_1$-$TWGM_4$ generation, visualized via $u_z$ distributions in Fig. 3d. Distance $D$ = 3.9–4.5$\lambda$ (relative to TES wavelengths at frequencies corresponding to $TWGM_2$-$TWGM_4$) confirms a remarkable far-field coupling. The anticlockwise energy flow is revealed in Fig. 3d, matching experimental Supplementary Movie S1. The $TWGM_1$ generation at the features labelled $P_1$ and $D_1$ is perturbed by bulk waves in the pure PnC; this perturbation is confirmed experimentally, as shown in Supplementary Fig. 9.

The inset of Fig. 3c displays zoomed-in numerical (dashed) and theoretical (solid) outlet $|u_z|$ profiles over the minima $D_2$- $D_4$, with the theoretical curves obtained from the dynamical equations (see Methods). Radiative loss $\gamma$ exceeds dissipative loss $\Gamma$ (the over-coupled regime) for all TWGMs owing to Dirac-strip enhancement. Material loss ($\alpha$ = 0.0025; see Methods) moderately affects the coupling associated with the minima labelled $D_3$ and $D_4$ (Supplementary Fig. 9), but significantly weakens the features associated with $D_1$ and $D_2$, whereas the cavity peak frequencies ($P_1$-$P_4$) remain stable. Besides material loss and the influence of bulk waves, the experiment is also affected by backward scattering[32] (arising from cavity corners[20]) and boundary effects at the metamaterial–silicon interface, whereas the input waves are limited in burst length (number of cycles) and pulse power (see Methods). These factors can prevent waves from fully propagating along the ring path as in idealized simulations. While the full demonstration of wave fields relies on simulations, experiments remain essential to reveal the functional capabilities of the waveguide circuits, serving as a foundation for future applications. Far-field phonon coupling enables

long-distance cavity energy localization and unlocks functionalities in complex systems, such as dual-cavity configurations.

## Single-cavity near-field waveguide system
Our second on-chip waveguide–cavity system integrates an NFC positioned above the waveguide and coupled via a Dirac strip of length $D$=3 × 0.866$a$ (Supplementary Fig. 10), which enables observation of near-field coupling. For this NFC-waveguide circuit, $TWGM_3$ gives rise to dip $D_3$. Experimentally, we position the NFC above the Dirac-strip-bridged waveguide. Despite geometric variations compared to the FFC system, $D_3$ and $P_3$ persist in experiment and simulation (Supplementary Fig. 11) regardless of cavity position. TWGM frequencies shift with changes in $s_l$ and $h_p$, but the robustness of the $D_3$ feature confirms the existence of stable coupling. Crucially, the NFC exhibits a 10 kHz frequency shift between the $D_3$ dip (1.955 MHz) and the corresponding $P_3$ peak (1.965 MHz), which is not observed in the FFC systems.

## Dual-cavity waveguide system
Our final waveguide-cavity phonon system integrates both FFC and NFC, bridged to a shared waveguide via two Dirac strips (Fig. 4a). The cavities are positioned on opposite sides to prevent direct cavity-to-cavity coupling. With parameters matching the single NFC circuit parameters, we measure $u_z$ to obtain the normalized $|u_z|$ at key positions (Fig. 4a). Focusing on $TWGM_3$, which generates transmission dips in individual cavities, the top panel of Fig. 4b shows peak values in normalized $|u_z|$ of 0.66 (NFC) and 0.34 (FFC) at 1.956 MHz (the FFC $TWGM_3$ frequency), and corresponding values of 0.53 (NFC) and 0.47 (FFC) at 1.973 MHz (the NFC $TWGM_3$ frequency). This confirms sympathetic resonances[33] at each $TWGM_3$ frequency−i.e., both cavities are resonating simultaneously−validated by the $|u_z|$ field maps (middle and bottom panels in Fig. 4b). Wave energy is thus exchanged between cavities. The time-resolved $u_z$ map (Supplementary Movie S2) reveals clockwise circulation in the upper cavity and anticlockwise circulation in the lower cavity.

In the top panel of Fig. 4b, the outlet $|u_z|$ shows a dip $D_{F3}$ at 1.956 MHz. Isolated FFC measurements confirm that the $D_{F3}$ dip originates from the resonance of the $TWGM_3$ in the FFC, whereas resonance hybridization in the dual-cavity system additionally

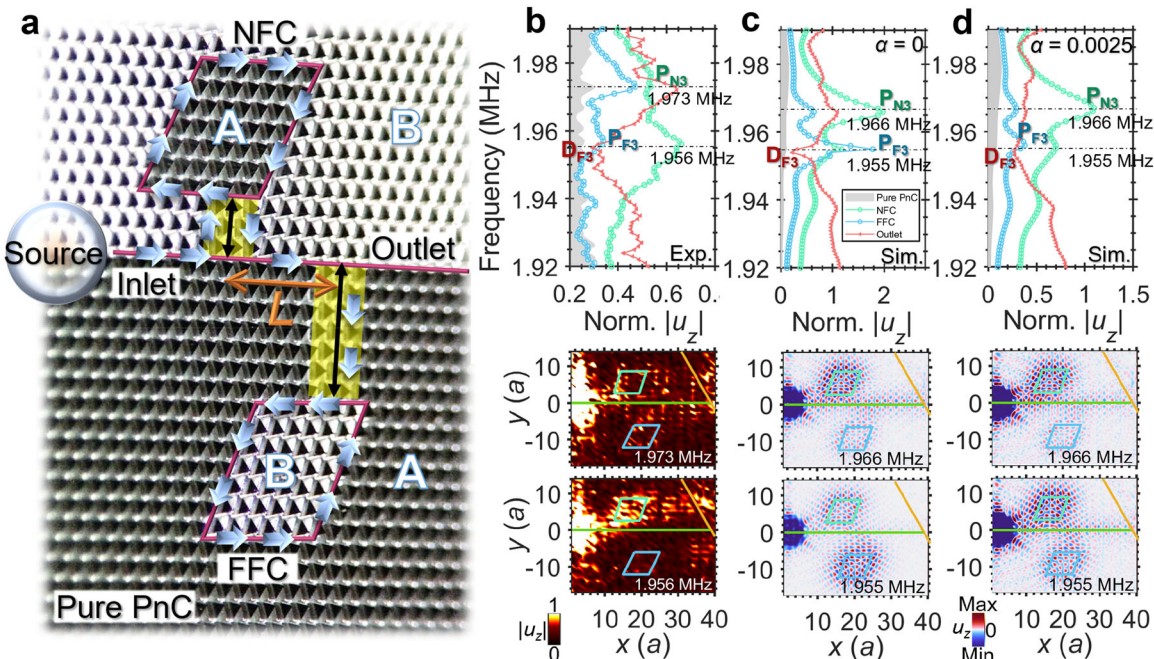

**Fig. 4 | Enhanced coupling and sympathetic resonances in the dual-cavity waveguide system. a** Optical micrograph of the dual-cavity waveguide phonon system with near-field cavity (NFC, $D = 3 \times 0.866a$) and far-field cavity (FFC, $D = 7 \times 0.866a$) coupled to the same waveguide via Dirac strips. The inter-cavity distance $L = 7a$. **b** Top: Measured normalized $|u_z|$ at the Pure PnC, NFC, FFC, and Outlet positions, excited by a circular pulsed source centered at $f_c = 1.9$ MHz. Bottom: $|u_z|$ distributions at 1.956 MHz ($P_{F3}$) and 1.973 MHz ($P_{N3}$). (F/N denotes FFC/NFC; $i$ is the peak/dip index.) The thin lines mark waveguide boundaries and cavity paths. **c, d** Simulated spectra corresponding to (**b**) obtained with material loss $\alpha = 0$ (**c**) and $\alpha = 0.0025$ (**d**).

couples $D_{F3}$ to the TWGM$_3$ in the NFC. This overlap explains why the outlet minimum labelled $D_{F3}$ reaches a normalized $|u_z| \approx 0.29$ here—slightly lower than the corresponding value of approximately 0.32 in the single FFC systems (simulation: 0.22 vs. 0.26). Energy concentrates primarily in the NFC (Fig. 4b bottom panel), which exhibits broad-spectrum $|u_z|$ profiles in Supplementary Fig. 12. The numerically computed fields at 1.955 MHz (DF$_3$, top panel in Fig. 4c) reveal a strong FFC $|u_z|$ (bottom panel), differing slightly from Fig. 4b. Introducing material loss $\alpha$ (Fig. 4d top panel), peaks $P_{F3}$ (1.955 MHz) and $P_{N3}$ (1.966 MHz) persist (Fig. 4d middle and bottom panels). Supplementary Figs. 13–14 show displacements versus $\alpha$, with peaks in $|u_z|$ at the features labelled $P_{N1}$–$P_{N4}$ and $P_{F1}$–$P_{F4}$ decreasing with increasing $\alpha$, whereas dips such as $D_{F2}$ and $D_{F4}$ exhibit V-shaped profiles with distinct minima.

Besides material loss, bulk waves are visible beyond the cavities in the wave field of Fig. 4b. As with the FFC circuit, experimental characterization of the full wave field along the ring paths is limited by our ultrasonic pulses of finite duration and power. Simulations capture the ideal wave-field behaviour, whereas experiments remain essential to reveal the main factors influencing wave propagation.

### Space-controlled sympathetic resonances

We now investigate space-controlled sympathetic resonances in the dual-cavity waveguide system by modulating the inter-cavity coupling through adjustment of the horizontal separation $L$ from $-7a$ to $16a$, probing how this coupling affects transmission dips. The simulated normalized $|u_z|$ at three positions is displayed in Fig. 5a: waveguide Outlet (left), NFC (middle), and FFC (right), without material loss. Four bright bars labeled $P_{N1}$-$P_{N4}$ emerge in the NFC panel, whereas four bars $P_{F1}$-$P_{F4}$ appear in the FFC panel. Frequencies for $P_{N1}$-$P_{N4}$ and $P_{F1}$-$P_{F4}$ match their isolated cavity counterparts.

Energy exchange between cavities is evidenced by fainter bright bars in the NFC at the frequencies corresponding to $P_{F1}$-$P_{F4}$ (e.g., dashed lines at $P_{F2}$ and $P_{F3}$) and in the FFC at the frequencies corresponding to $P_{N1}$-$P_{N4}$. The outlet panel shows dark bars labelled $D_{F1}$-$D_{F4}$

and $D_{N4}$, corresponding to the resonances $P_{F1}$-$P_{F4}$ and $P_{N4}$, respectively.

We now analyze the dual-cavity energy exchange mechanism using governing equations of motion. Let $P_{Fj}'$ denote denote the peaks and $D_{Fj}'$ denote the outlet dips for the single cavity circuit, and let $P_{Fj}''$ and $D_{Fj}''$ denote the corresponding quantities for the dual-cavity circuit ($j = 1,2,3,4$). Without material losses, the outlet dips in the dual-cavity circuit are given by $D_{Fj}'' = D_{Fj}' \times P_{Fj}'/P_{Fj}''\left(1 + P_{FNj}''\right)$ (see theory). Based on this formula, theoretically predicted results (square markers) for $D_{F2}$-$D_{F4}$ are consistent with direct simulations (solid lines) from the dual-cavity model (Fig. 5b). Relevant results for $D_{F1}$ appear in Supplementary Fig. 15.

In the expression for $D_{Fj}''$, the term $P_{FNj}''$ ($j = 1, 2, 3, 4$) in the denominator refers to the inter-cavity coupling, and serves to reduce the value of $D_{Fj}'$. The response of this system therefore cannot be expressed as a linear superposition of two independent cavities. When $L$ is close to 0, the value of $P_{Fj}''$ is markedly reduced, e.g., for $P_{F1}$-$P_{F3}$, which leads to a large value of $D_{Fj}''$ for $D_{F1}$-$D_{F3}$, so that critical coupling no longer occurs. Enhanced transmission is not the sole functionality accessible in dual-cavity systems; alternative structural designs can be used to target different wave-control objectives, such as band-gap broadening achieved with double-sided pillars[34]. Here, the small value of $P_{Fj}''$ when $L \approx 0$ reflects valley PnC properties, such as valley-selected routing paths[24].

The energy exchange between the dual cavities demonstrated here is not only crucial for the emergence of non-Hermitian phenomena, but also underpins sympathetic resonance at shared frequencies. Such non-Hermitian behaviour is vital for high-performance sensing and information processing.

### Discussion

In conclusion, we have presented an experimental realization of a far-field phonon coupling circuit by means of a Dirac-strip valley metamaterial. The Dirac strip acts as a broadband, direction-selective transport channel. The surrounding band-gap phononic crystals

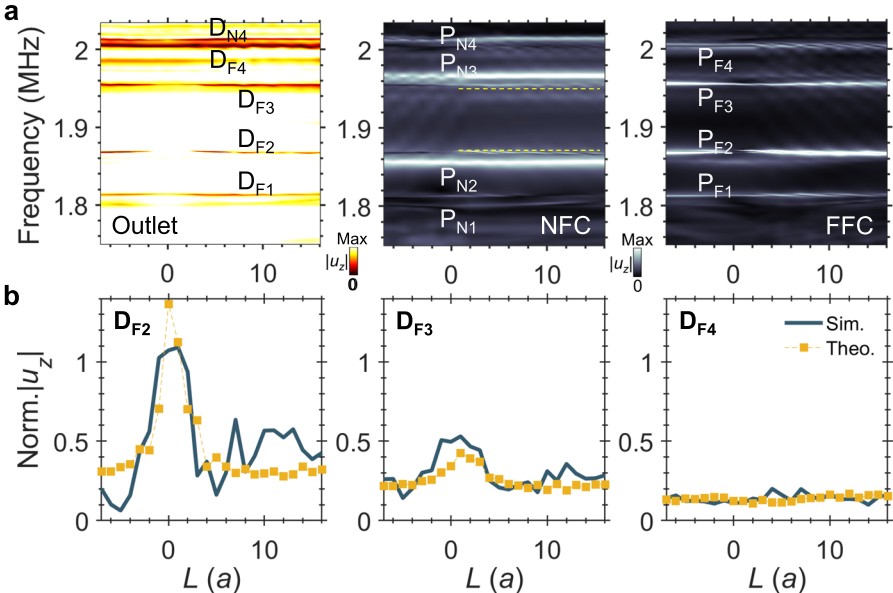

**Fig. 5 | Space-controlled coupling and sympathetic resonances in the dual-cavity waveguide system. a** Computed normalized $|u_z|$ plotted against frequency and inter-cavity distance $L$ at (left) the Outlet, (middle) the upper edge of the NFC, and (right) the bottom edge of the FFC (positions as in Fig. 4a). Positive $L$ indicates suppress lateral energy spreading, and the linear Dirac dispersion enables phase-coherent transport over multiple wavelengths, providing a quasi-one-dimensional pathway that efficiently guides cavity radiation toward the waveguide and back. This simultaneously achieves robustness, spectral multiplexing, and distance-insensitive waveguide-cavity energy transfer in an on-chip configuration. In addition, by integrating cavities operating in the near and far fields, we establish deterministic control over non-Hermitian dynamics through material loss and spatial separation, experimentally observing sympathetic resonances.

NFC is to the left of FFC; negative $L < 0$ indicates NFC to the right of FFC. **b** Three plots of normalized $|u_z|$ vs. $L$ extracted from $D_{F2}$ to $D_{F4}$ in (**a**), compared with theoretical results calculated from $D_{Fj'} \times P_{Fj'}/P_{Fj''}\left(1 + P_{FNj''}\right)$.

In the single-cavity waveguide system, adjusting the length of the Dirac strip optimizes the far-field phonon coupling. In the dual-cavity waveguide system, the simultaneous excitation of both cavities produces a pronounced outlet dip compared to the single-cavity system, providing a clear signature of sympathetic cavity coupling. Furthermore, tuning the inter-cavity distance allows modulation of energy exchange between the dual cavities. The demonstration of far-field coupling over distances up to ~$5\lambda$ resolves the fundamental challenge of broadband coupling at multiwavelength distances—with direct implications for continuum-mechanical systems requiring spatial decoupling, quantum platforms demanding protected state transfer, photonics, and scalable phononic processors. Furthermore, our platform enables unprecedented pathways for nonlinear wave dynamics[33], analogs of quantum many-body phenomena[35], and topological wave engineering[36,37], directly addressing the escalating need for multiphysical control in integrated wave technologies.

## Methods

### Experimental methods

The phononic crystal samples were fabricated on 525 μm-thick silicon wafers using lithography and dry etching techniques. During lithography, a 6 μm photoresist layer was applied as an etching mask to protect unpatterned regions, followed by silicon etching in an LPX-ICP system using a BOSCH process to achieve vertical sidewalls. The BOSCH process alternated between passivation and etching steps, with parameters detailed in Supplementary Table 2. Pillar height control was achieved by timing the etching duration based on pre-calibrated silicon etch rates, and final geometries were optically verified.

For all samples (Fig. 1), the wave propagation was characterized using piezo-laser ultrasonics for in-situ time-resolved measurement of out-of-plane displacement (Supplementary Fig. 1). The displacement $u_z$ was monitored across a $41a \times 29a$ rectangular region encompassing cavities and waveguides, with spatial sampling intervals of 641 μm ($a$) along $x$ and 555 μm (0.866$a$) along $y$. Time-resolved data enabled derivation of average amplitude at the Inlet, Outlet, FFC, NFC, and Pure PnC regions (Figs. 3a and 4b, Supplementary Figs. 11a and 12a). In addition, we carried out spatial field mapping of $|u_z|$ at the resonant frequencies (Figs. 3b and 4b, Supplementary Figs. 9d, 11b and 12b).

As shown in Supplementary Fig. 1, the experimental setup comprised a Polytec OFV 2570 laser Doppler vibrometer (LDV), a high-speed camera, a RIGOL DG1032z signal generator, a power amplifier, and a Tektronix DPO4102B-L oscilloscope. The samples were mounted on a supporting plate with a spatial positioning precision of ~5 μm. A 5 mm × 1 mm PZT disk was bonded to the back surface of the waveguide inlet using conductive adhesive. A seven-cycle sinusoidal burst signal centered at frequency $f_c$ was generated, amplified, and delivered to the PZT transducer. The number of cycles in the burst was limited to facilitate identification of the incident wave packet as it propagated through the valley metamaterial phononic circuits. The LDV measured the out-of-plane displacement ($u_z$) within target regions, while the high-speed camera recorded the laser beam position and enabled spatial point selection.

To enhance experimental accuracy, several measures were taken. The input signal power was amplified by ~37 dB to generate ultrasonic pulses with sufficient amplitude for reliable measurement, without damaging the silicon wafer. At each spatial point, 512 scans were averaged to reduce white noise. The sampling frequency was set to $f_s = 100$ MHz, more than 47 times the target frequency $f_c$ (1.7–2.1 MHz). At each spatial point, signals were recorded with $10^5$ data points, corresponding to >1700 periods of $1/f_c$. This allowed fine spectral filtering near $f_c$ with a frequency resolution of ~0.001 MHz. By repeating these procedures, we obtained the amplitude of the out-of-plane displacement across the target region, enabling enabling clear resolution of neighboring TWGM peaks and adjacent dips. These procedures also allowed us to reconstruct animations and spatial segments of wave displacement within the region of interest.

## Numerical calculations

We adopt the finite-element method to calculate dispersion in our silicon phononic crystal plates. Elastic constants are taken as $C_{11} = 165.6$ GPa, $C_{12} = 63.9$ GPa, $C_{44} = 79.5$ GPa, and mass density 2.331 g/cm$^3$[38], with elastic anisotropy fully included; however, the anisotropy does not affect the qualitative PnC design principles. Material loss is introduced in the simulations by assigning a small imaginary part to the elastic constants $C_{ij}$ through a dimensionless loss parameter a, such that $C_{ij} \rightarrow C_{ij}(1+ia)$. Unit cell dispersion is computed for rotation angles $\theta = 0°$ and $\pm 20°$ (Fig. 1c, d).

For topological edge states (TESs), a sandwiched supercell (Supplementary Fig. 5a) comprising a 12-layer PnC-B (top), a 12-layer PnC-A (middle), and a 12-layer PnC-B (bottom) is constructed. Periodic boundary conditions are applied to the supercell B-A-B along the $x$-direction, whereas continuity conditions are applied along the $y$-direction. The results for dispersion (Supplementary Fig. 5b, c) include the displacement ratio $\int |u_z| dV / \int |u_t| dV$ at each frequency, where $u_z$ and $u_t$ denote out-of-plane and total displacements.

Topological whispering gallery modes (TWGMs) are analyzed using a rhombus cavity supercell (side length $7a$) positioned between the inner PnC-B and outer PnC-A (Supplementary Fig. 6a). Periodic boundary conditions are applied to three pairs of parallel sides of the hexagonal supercell, yielding the corresponding eigenfrequencies and displacement fields (Fig. 2b; Supplementary Fig. 6b, c).

Wave propagation simulations for the NFC-waveguide, FFC-waveguide (Fig. 2a), and dual-cavity structures (Fig. 4a) replicate experimental dimensions (Fig. 1a). Perfectly matched layers (PMLs) absorb boundary reflections. A circular source applies unit $z$-axis force over a 2.5 mm radius at the waveguide inlet. Transmission spectra (Figs. 3c and 4c, d, Supplementary Figs. 9a, 11c, 12c and 13) are based on the average $|u_z|$ over $5a$ segments at Inlet, Outlet, FFC, NFC, and unpatterned PnC regions.

In single cavity-waveguide models (Fig. 2a), the Dirac strip length $D$ varies from 1 to 12 layers in 1-layer increments. Frequency- and distance-resolved $|u_z|$ maps (Fig. 2c) are generated from average values over $5a$ segments at the Outlet and FFC. Control simulations replacing Dirac strips with PnC-A appear in Fig. 2d. For dual-cavity models (Fig. 4a), the inter-cavity distance $L$ sweeps from $-7a$ to $16a$ in steps of $a$, producing frequency-distance $|u_z|$ maps at the Outlet, NFC, and FFC (Fig. 5a).

## Theory of the waveguide-cavity dynamics

### Single-cavity waveguide system.

The waveguide-cavity coupling can be understood from the theoretical model shown in Supplementary Fig. 2. The cavity lies either above or below the waveguide. From theory[39,40], the incident wave $S_+$ gives rise to a cavity response $\tilde{a}_c = a_c e^{i\omega t}$. $\Gamma$ and $\gamma$ are the dissipative and radiative losses of the cavity, respectively.

The governing equation can be expressed as[40–42]

$$\frac{d\tilde{a}_c}{dt} = (i\omega_0 - \Gamma - \gamma)\tilde{a}_c + i\sqrt{\gamma}S_+ , \qquad (1)$$

where $\omega_0 = 2\pi f_0$ represents the resonant frequency of the cavity. Silicon can be considered to be close to an isotropic material, so, in contrast to previous work[20], material anisotropy is ignored in this equation.

The derivative of $\tilde{a}_c = a_c e^{i\omega t}$ is given by

$$\frac{d\tilde{a}_c}{dt} = \frac{d(a_c e^{i\omega t})}{dt} = \frac{da_c}{dt}e^{i\omega t} + a_c \frac{d(e^{i\omega t})}{dt} = \frac{da_c}{dt}e^{i\omega t} + i\omega\tilde{a}_c. \qquad (2)$$

When the amplitude remains constant, $da_c/dt = 0$, yielding

$$\frac{d\tilde{a}_c}{dt} = i\omega\tilde{a}_c. \qquad (3)$$

Substituting Eq. (3) into Eq. (1) yields

$$\tilde{a}_c = \frac{i\sqrt{\gamma}}{i(\omega - \omega_0) + \Gamma + \gamma}S_+ . \qquad (4)$$

The relationship between the input wave and the output wave $S_-$ can be written as

$$S_- = S_+ + i\sqrt{\gamma}\tilde{a}_c. \qquad (5)$$

Combining Eqs. (4) and (5), the transmission coefficient of the system can be expressed in the form

$$T = \left|\frac{S_-}{S_+}\right| = \left|\frac{S_+ + i\sqrt{\gamma}\tilde{a}_c}{S_+}\right| = \left|1 + \frac{-\gamma}{i(\omega - \omega_0) + \Gamma + \gamma}\right|. \qquad (6)$$

According to Eq. (6), one obtains

$$\begin{cases} \Delta\omega = 2(\Gamma + \gamma), \\ T_{\min} = \frac{\Gamma}{\Gamma + \gamma}, \end{cases} \qquad (7)$$

where $T_{\min}$ denotes the minimum value around $\omega_0$, and $\Delta\omega$ is the angular frequency at full width at half maximum of the corresponding transmission dip. These two parameters can be obtained from the numerically calculated spectrum. The substitution of $T_{\min}$ and $\Delta\omega$ into Eq. (7) yields the dissipative loss $\Gamma$ and radiation loss $\gamma$. Then, by use of Eq. (6), one can plot the fitted transmission profile around $\omega_0$. Critical coupling occurs when $\Gamma = \gamma$, whereas $\Gamma < \gamma$ (or $\Gamma > \gamma$) indicates over- (or under-) coupling.

### Dual-cavity waveguide system.

The theoretical model of the dual-cavity waveguide system is illustrated in Supplementary Fig. 3. Each cavity includes dissipative loss $\Gamma_j$ and radiative loss $\gamma_j$ ($j = F, N$). Since the two cavities are inserted on separate sides of the waveguide, direct cavity-to-cavity coupling is prohibited. The quantity $\varphi$ is the accumulated phase difference along the waveguide path.

For an incident wave $S_+$, the governing equation for the two-cavity system ($\tilde{a}_N = a_N e^{i\omega t}$ and $\tilde{a}_F = a_F e^{i\omega t}$) can be written as[43,44]

$$\frac{d\tilde{a}_N}{dt} = (i\omega_N - \Gamma_N - \gamma_N)\tilde{a}_N + i\sqrt{\gamma_N}(S_+ + i\sqrt{\gamma_F}e^{-i\varphi}\tilde{a}_F), \qquad (8)$$

$$\frac{d\tilde{a}_F}{dt} = (i\omega_F - \Gamma_F - \gamma_F)\tilde{a}_F + i\sqrt{\gamma_F}(S_+ e^{-i\varphi} + i\sqrt{\gamma_N}e^{-i\varphi}\tilde{a}_N), \qquad (9)$$

where $\omega_N$ and $\omega_F$ are the resonance frequencies of the near- and far-field cavities, respectively. Direct cavity-to-cavity coupling is ignored. On the right-hand side of Eq. (8), the 2$^{\text{nd}}$ term includes two contributions: one from the coupling between the upper cavity ($a_N$) and the incoming waves, the other from the radiated field from the lower cavity ($a_F$). Equation (9) can be understood in a similar way.

The transmission coefficient for the system is given by

$$T = \left|\frac{S_-}{S_+}\right| = \left|\frac{(S_+ + i\sqrt{\gamma_N}\tilde{a}_N)e^{-i\varphi} + i\sqrt{\gamma_F}\tilde{a}_F}{S_+}\right| \\ = \left|e^{-i\varphi} + i(e^{-i\varphi}\sqrt{\gamma_N}\tilde{a}_N/S_+ + \sqrt{\gamma_F}\tilde{a}_F/S_+)\right|. \qquad (10)$$

By fitting the simulated transmission profile, we extract the parameters $\Gamma_j$ and $\gamma_j$ ($j = F, N$). Substituting them into Eqs. (8) and (9) yields $\tilde{a}_N$ and $\tilde{a}_F$. One can then plot the theoretically fitted profile of transmission from Eq. (10).

When $\omega = \omega_F$,

$$T = \left| e^{-i\varphi} + i(e^{-i\varphi}\sqrt{\gamma_N}a_N/S_+ + \sqrt{\gamma_F}a_F/S_+) \right| = D_{Fj}''. \tag{11}$$

Specifically,

$$D_{Fj}'' = \left| \frac{e^{i\varphi}([i(\omega - \omega_F) + \Gamma_F + \gamma_F] - \gamma_F)([i(\omega - \omega_N) + \Gamma_N + \gamma_N] - \gamma_N)}{e^{2i\varphi}[i(\omega - \omega_N) + \Gamma_N + \gamma_N][i(\omega - \omega_F) + \Gamma_F + \gamma_F] - \gamma_F\gamma_N} \right|, \tag{12}$$

where the $D_{Fj}''$ is the outlet dip in the dual-cavity circuit.

Combining Eqs. (8) and (9), the amplitudes for the NFC and FFC are given by

$$\left| \frac{\tilde{a}_N}{S_+} \right| = \left| \frac{i\sqrt{\gamma_N}\{e^{2i\varphi}[i(\omega - \omega_F) + \Gamma_F + \gamma_F] - \gamma_F\}}{e^{2i\varphi}[i(\omega - \omega_N) + \Gamma_N + \gamma_N][i(\omega - \omega_F) + \Gamma_F + \gamma_F] - \gamma_F\gamma_N} \right|, \tag{13}$$

$$\left| \frac{\tilde{a}_F}{S_+} \right| = \left| \frac{i\sqrt{\gamma_F}\{e^{i\varphi}[i(\omega - \omega_F) + \Gamma_F + \gamma_F] - e^{i\varphi}\gamma_N\}}{e^{2i\varphi}[i(\omega - \omega_N) + \Gamma_N + \gamma_N][i(\omega - \omega_F) + \Gamma_F + \gamma_F] - \gamma_F\gamma_N} \right| = P_{Fj}'', \tag{14}$$

where $P_{Fj}''$ refers to the amplitude peak for the FFC ($\omega = \omega_F$) in the dual-cavity circuit.

As a comparison, consider again the single FFC-waveguide model and set $\omega = \omega_0 = \omega_F$. According to Eqs. (4) and (6), the outlet dip $D_{Fj}'$ and the FFC peak $P_{Fj}'$ can be derived as follows:

$$D_{Fj}' = \left| 1 - \frac{\gamma_F}{\Gamma_F + \gamma_F} \right|, \tag{15}$$

$$P_{Fj}' = \left| \frac{i\sqrt{\gamma_F}}{\Gamma_F + \gamma_F} \right|. \tag{16}$$

Combining Eqs. (12), (14), (15) and (16),

$$\frac{D_{Fj}''P_{Fj}''}{D_{Fj}'P_{Fj}'} = \left| \frac{e^{2i\varphi}[i(\omega - \omega_F) + \Gamma_F + \gamma_F]^2\{[i(\omega - \omega_N) + \Gamma_N + \gamma_N] - \gamma_N\}^2}{\{e^{2i\varphi}[i(\omega - \omega_N) + \Gamma_N + \gamma_N][i(\omega - \omega_F) + \Gamma_F + \gamma_F] - \gamma_F\gamma_N\}^2} \right|. \tag{17}$$

If the two cavities possess different resonance frequencies ($\omega_N \neq \omega_F$), then when $\omega = \omega_F$,

$$i(\omega - \omega_F) + \Gamma_F + \gamma_F|_{\omega = \omega_F} = \Gamma_F + \gamma_F, \tag{18}$$

and the term $i(\omega - \omega_N) + \Gamma_N + \gamma_N$ can be seen to contain extra terms compared to $\Gamma_F + \gamma_F$:

$$i(\omega - \omega_N) + \Gamma_N + \gamma_N = \Gamma_F + d\Gamma + \gamma_F + d\gamma + id\omega. \tag{19}$$

One can rewrite Eq. (17) as

$$\frac{D_{Fj}'P_{Fj}'}{D_{Fj}'P_{Fj}''} = \left| \frac{e^{-2i\varphi}(\gamma_F^2 - e^{2i\varphi}(\gamma_F + \Gamma_F)^2)^2}{\Gamma_F^2(\gamma_F + \Gamma_F)^2} \right| + O(d\Gamma, d\gamma, d\omega), \tag{20}$$

and similarly for Eq. (20):

$$\frac{D_{Fj}'P_{Fj}'}{D_{Fj}''P_{Fj}''} = \frac{(2\gamma_F + \Gamma_F)^2}{(\gamma_F + \Gamma_F)^2} + O(d\Gamma, d\gamma, d\omega, \varphi) = 1 + P_{FNj}'', \tag{21}$$

where

$$P_{FNj}'' = \frac{2\gamma_F}{\gamma_F + \Gamma_F} + \left( \frac{\gamma_F}{\gamma_F + \Gamma_F} \right)^2 + O(d\Gamma, d\gamma, d\omega, \varphi). \tag{22}$$

Finally, the outlet dip in the dual-cavity circuit can be expressed as

$$D_{Fj}'' = D_{Fj}' \frac{P_{Fj}'}{P_{Fj}''(1 + P_{FNj}'')}. \tag{23}$$

Based on Eq. (12), the outlet dip is governed by $\Gamma_j$ and $\gamma_j$ ($j = F, N$). Equation (23) shows that the inter-cavity coupling affects the outlet dip $D_{Fj}''$, which can be seen in the term $(1 + P_{FNj}'')$.

## Data availability

The data that support the findings of this study are available from the corresponding author upon request.

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

## Acknowledgements

This work was supported by the National Natural Science Foundation of China (Grants No. 12172256, 12302115, 12272269, 11972257, 52027816). It was also supported by Shanghai Leading Talent Program of Eastern Talent Plan. J. Z. would like to acknowledge Dr. Haoyun Tu's help with the experiment.

## Author contributions

Yao H., Weitao Y. and Jinfeng Z. designed the sample structures. Qi W. and Jia Z. fabricated the experimental samples. Yao H. performed the experimental measurements, carried out the numerical simulations, analyzed and visualized all data, and wrote the first draft of the manuscript. Weitao Y. and Jinfeng Z. helped check the simulation results. Weitao Y., Yao H., Zhiwei G. and Jinfeng Z. derived the theory. Jinfeng Z. conceived the idea, designed the experimental setups and analyzed the initial experimental data. Yuxuan Z. and Yongdong P. assisted in the experiments. Yueting Z., Zheng Z. and Oliver B. Wright helped in the derivation of the theory. Jinfeng. Z., Weitao Y., Yueting Z., Yongdong P. and Zhiwei G. conceived the project. Oliver B. Wright constructed the final version of the manuscript. All the authors contributed to discussions, interpretation of the data, and the writing process.

## Competing interests

The authors declare no competing interests.
