## [Peer Review File · Nature Communications]

Far-field phonon coupling in valley metamaterial circuits

Corresponding Author: Professor Jinfeng Zhao

Version 0:

Reviewer comments:

Reviewer #1

(Remarks to the Author)

This manuscript presents the topological and non-Hermitian physics in on-chip waveguide-cavity systems, addressing challenges in far-field phonon coupling and multi-cavity dynamics. The authors introduce broadband and robust "Dirac strip" to conquer far-field coupling in valley topological systems. This is experimentally validated through observations of displacement peaks or dips via pump-probe interferometry, demonstrating robust phonon coupling in single-cavity system. The dual-cavity system further reveals non-Hermitian behaviors, where material loss and inter-cavity spacing modulate synchronizing oscillations—findings supported by a theory of dual-cavity energy exchange. The experimental technique is quite smart, while the theory underlines well the influences of observed synchronizing oscillations.

Given the rigorous experiments, theoretical insights, and broad interdisciplinary significance (phonon, photon, and topological metamaterials), I find this work quite suitable for publication in Nature Communications. I strongly recommend its publication after addressing the following issues:

1. The triangular pillar serves as the unit cell in this work. However, the motivation of this selected specific geometry was not elucidated in the main text or supplementary information. Why choose this type of unit cell as the fundamental configuration?
2. In the last paragraph of the Introduction, the second sentence is ambiguous. What does the authors want to express about "in range and efficiency"? I suggest modifying this sentence to make it clearer.
3. In the first paragraph of Results Section, the authors notify the Supplementary Figs. 5-7 in one place. This is not clear. The explicit linking between Supplementary Figs. 5-7 to relevant sentences in this paragraph shall be given.
4. After Fig. 1, clear description is needed about the part "The modeled antisymmetric bulk wave..... shaping the D1-D4 and P1-P4 $|uz|$ profiles". Additionally, is it possible to add a displacement field at strip pillar angle $\theta = 10^\circ$ as an example in Supplementary Figure 10?
5. While the key achievements are highlighted in the Discussion, more descriptions on detailed findings are needed to provide crucial guidance for further research, e.g., the feasible layers of Dirac strip, conditions for stronger critical coupling, etc.
6. The authors enable the far-field phonon coupling and present the non-Hermitian features such as synchronizing oscillations. Why these two are important or essential needs highlight.
7. The authors should present numerical methods to calculate the eigenfrequencies of supercells for TESs and TWGMs, e.g., B-A-B elongated and B-inner A-outer hexagonal supercells. Additionally, it should be clarified whether periodic boundary conditions are applied.
8. How the authors distinguish the transmission peaks or dips in experiment? As we can see the frequency difference between the two peak frequencies in Fig. 3b-d is only about 0.01 MHz versus the central frequency 1.96 MHz.

Reviewer #2

(Remarks to the Author)

The noteworthy result presented in the manuscript is the demonstration of 'far-field' coupling between a waveguide and cavities in a valley phononic plate crystal, through a 'Dirac strip'. Among the claims, I was not convinced by what the authors call 'synchronized oscillations': in the understanding of this reviewer there is no oscillator in the experiments presented, but resonators excited by a single-frequency driving source (the term 'oscillator' has a clear meaning in physics and engineering and appears to be misused). The discussion of non-Hermitian circuits also sounds off for the same reason: there is dissipative (material, structure) loss and radiation loss, for sure, but there is no gain. The Methods section says that 'direct cavity-to-cavity coupling is prohibited', then why does the main text advertises 'synchronized oscillations in the resonators',

whereas clearly the same external source drives both of them?

Strangely, the supplemental information is much clearer and informative than the main manuscript. It is written in simple engineering terms and the figures are clear and not composed of too many panels. What the main manuscript calls 'pump-probe' is actually a piezoelectric source and a laser vibrometer, there is no laser pump (while first reading the text I thought laser ultrasound was being used in the experiment but this not the case at all).

Globally the methodology is consistent, but most of the useful information is given in the SI, such as geometric parameters, band gap frequencies, Dirac cone deformation with angle θ - I personally tend to dislike this way of writing papers, aiming at impact rather than accuracy. I recommend a reorganization of the manuscript with important figures of the SI included in the main text (SI figures 1, 2, 5 typically). The revision should be made in order to clarify the methodology and to reconsider the claims: there is no reason to oversell what has been achieved, the experiment is rather nice and seems well conducted. As a final note, the experimental results in figures 2 and especially figure 3 are not really successful in demonstrating localized fields established inside the cavities; whatever the experimental reasons it seems the conclusions can only rely on numerical simulations - this should be acknowledged.

Version 1:

Reviewer comments:

Reviewer #1

(Remarks to the Author)

In the respond letter, authors point to point response to my comments about the details of this study. After thorough-going analysis and clear description the details of the study, authors reasonably show the significance of this work. In a word, this paper is a careful study and the findings are of considerable interest. This paper is worth to be published.

Reviewer #2

(Remarks to the Author)

The authors have conducted a very convincing revision. I am highly satisfied with all answers to my comments and suggestions. The correction of terminology, I hope, makes the manuscript clearer regarding the precise achievements. The redistribution of information between main text and supplemental text is now adequate.

Overall this is a very well written manuscript (I did not spot any typo), concise and precise in the methodology. The conclusions are well supported by the results. I believe the manuscript is ready for publication and I recommend acceptance.

Point-by-point response

Dear Editor,

We would like to thank you and the reviewers for the quick response. We also thank the Reviewers for the recognition of this work and for their valuable suggestions. We greatly appreciate the opportunity to revise this work. All concerns and recommendations have been carefully addressed to enhance the quality of our manuscript. The point-by-point responses are highlighted by the red text in the manuscript. Our responses are given below.

Response to reviewer #1

Reviewer #1: This manuscript presents the topological and non-Hermitian physics in on-chip waveguide-cavity systems, addressing challenges in far-field phonon coupling and multi-cavity dynamics. The authors introduce broadband and robust "Dirac strip" to conquer far-field coupling in valley topological systems. This is experimentally validated through observations of displacement peaks or dips via pump-probe interferometry, demonstrating robust phonon coupling in single-cavity system. The dual-cavity system further reveals non-Hermitian behaviors, where material loss and inter-cavity spacing modulate synchronizing oscillations—findings supported by a theory of dual-cavity energy exchange. The experimental technique is quite smart, while the theory underlines well the influences of observed synchronizing oscillations.

Given the rigorous experiments, theoretical insights, and broad interdisciplinary significance (phonon, photon, and topological metamaterials), I find this work quite suitable for publication in Nature Communications. I strongly recommend its publication after addressing the following issues:

Authors: We thank the referee for the positive comments. We have carefully revised both the main text and supplementary information, highlighting modifications in red.

Reviewer #1: 1. The triangular pillar serves as the unit cell in this work. However, the motivation of this selected specific geometry was not elucidated in the main text or supplementary information. Why choose this type of unit cell as the fundamental configuration?

Authors: Thank you for noting the choice of the triangular pillar as the unit cell. The motivation for selecting this specific geometry is rooted in its symmetry properties (C_{3v} symmetry when θ equals zero) and the potential to induce topological phase transitions. We adopted this classic valley PnC that support topological states, at the aim of exploring novel mechanisms of wave propagation.

Relevant revisions are made as follows:

Page 2 in the main text, “The unit cell of triangular pillars possesses C_{3v} symmetry when θ equals zero, and topological phase transitions are induced by changing rotation angle θ to break this symmetry. These valley PnCs support the topological states reported previously^{24,29}. We adopt this classic paradigm and fabricate all three samples on silicon wafers to explore novel mechanisms of wave propagation.”

Reviewer #1: 2. In the last paragraph of the Introduction, the second sentence is ambiguous. What does the authors want to express about "in range and efficiency"? I suggest modifying this sentence to make it clearer.

Authors: Thank you for this valuable suggestion. For the word "range", we mainly focus on the far-field phonon coupling over long distances by using the Dirac strip. For the word "efficient", we underline that the Dirac strip enhances the coupling efficient between the waveguide and the cavity.

Relevant revisions are made as follows:

Page 2 in the main text, "This is achieved through topological phonon interactions in on-chip valley waveguide-cavity systems using a Dirac-cone waveguide strip (a 'Dirac strip'), **which enables long-distance coupling beyond conventional near-field limits and significantly improves coupling efficiency.**"

Reviewer #1: 3. In the first paragraph of Results Section, the authors notify the Supplementary Figs. 5-7 in one place. This is not clear. The explicit linking between Supplementary Figs. 5-7 to relevant sentences in this paragraph shall be given.

Authors: Thanks for your kind reminder. We have addressed this part you mentioned to explicitly link each relevant sentence to the corresponding Supplementary Figs. 5-7 (now Figs. 1b-1d and Supplementary Figs. 3-4), in the main text (Pages 2-3).

Relevant revisions are made as follows:

Page 2 in the main text, "**Figure 1a shows the three valley metamaterial circuits investigated in this work. Sample 1 is a far-field cavity (FFC) waveguide circuit composed of triangular pillars, with side length $s_l = 500 \mu\text{m}$ and height $h_p = 292 \mu\text{m}$ (see Fig. 1b).**"

Page 2 in the main text, "**Dirac cones appear at the K point of the Brillouin zone for $\theta = 0^\circ$ (Figs. 1c-d, upper panels), while material anisotropy introduces a minor band gap (insets).**"

Page 3 in the main text, "PnCs A and B exhibit opposite topological phases (**Supplementary Fig. 3**), and their interface supports topological edge states (TESs, **Supplementary Fig. 4**) and topological whispering gallery modes (TWGMs, Fig. 2b and Supplementary Fig. 5)."

Reviewer #1: 4. After Fig. 1, clear description is needed about the part "The modeled antisymmetric bulk wave..... shaping the D1-D4 and P1-P4 $|uz|$ profiles". Additionally, is it possible to add a displacement field at strip pillar angle $\theta = 10^\circ$ as an example in Supplementary Figure 10?

Authors: Based on your first point, we have reorganized this part and provided a more detailed description, in the main text (Page 5). Besides, we note the influence of the bulk branch of the Dirac strip on cavity-waveguide coupling, in the Supplement (Page 7).

Regarding the second question, we have added a displacement field at strip pillar angle $\theta = 10^\circ$ for TWGM₄ in Supplementary Figure 10 (now Supplementary Figure 7a), in the Supplement (Page 6).

Relevant revisions are made as follows:

Page 5 in the main text, “The modeled antisymmetric bulk-wave branches emerge with wave energy confined to the Dirac strip, whereas their frequency ranges occupy the eigenfrequencies of TWGM₁, TWGM₃ and TWGM₄ (Supplementary Fig. 7). These guided bulk-wave branches are modulated by strip pillar rotation. For example, at strip pillar angle $\theta = 10^\circ$, these bulk-wave branches occupy exclusively the TWGM₄ frequency, directly shaping the D₁-D₄ and P₁-P₄ $|u_z|$ versus θ profiles (Supplementary Fig. 7).”

Page 6 in the Supplement, “

Supplementary Figure 7. The influence of pillar rotation in the PnC-C strip (based on the sample in Fig. 2a). **a** The FFC-waveguide structure and the u_z distribution when $\theta = 10^\circ$ for TWGM₄ (upper panel). Calculated normalized $|u_z|$ retrieved at the outlet (dip value, orange circle) and at the lower edge of the cavity (peak value, blue circle), versus the pillar rotation of the PnC-C in the model (upper panel). Dashed lines are fitted curves. PnC-C bridges the waveguide and the cavity, and becomes the Dirac strip when $\theta = 0$. **b** Supercell (upper panel) used to calculate the dispersion (lower left panel) when $\theta = 0^\circ$. The color scale indicates the ratio of u_z to the total displacement u_t in the supercell. The middle panel shows a map of u_z at 1.93 MHz when $\theta = 0^\circ$. Evolution of the eigenfrequencies (lower right panel) versus the pillar rotation for PnC-C when $k_y = 0$.”

Page 7 in the Supplement, “Thanks to the overlap between the bulk branch of the Dirac strip and the resonant frequency of the cavity for TWGM₃, highly-efficient matched coupling becomes possible between the horizontal waveguide and the far-field cavity.”

Reviewer #1: 5. While the key achievements are highlighted in the Discussion, more descriptions on detailed findings are needed to provide crucial guidance for further research, e.g., the feasible layers of Dirac strip, conditions for stronger critical coupling, etc.

Authors: Thanks for your advice about more detailed findings in the Discussion for further research. So, we have added related contents to enrich the findings in the main text (Page 9), i.e., in single-cavity systems, adjusting Dirac strip length optimizes far-field coupling. in dual-cavity systems, simultaneous cavity excitation creates a deeper outlet dip, and tuning inter-cavity distance modulates energy exchange.

Relevant revisions are made as follows:

Page 9 in the main text, “In the single-cavity waveguide system, adjusting the length of the Dirac strip optimizes the far-field phonon coupling. In the dual-cavity waveguide system, the simultaneous excitation of both cavities produces a pronounced outlet dip compared to the single-cavity system. Furthermore, tuning the inter-cavity distance allows modulation of energy exchange between the dual cavities.”

Reviewer #1: 6. The authors enable the far-field phonon coupling and present the non-Hermitian features such as synchronizing oscillations. Why these two are important or essential needs highlight.

Authors: Thanks for pointing out the need to highlight the importance of far-field phonon coupling and non-Hermitian feature.

First, the far-field phonon coupling is vital since it enables long-distance cavity energy localization. This is crucial for exploring functionalities in complex systems like dual cavity systems. We have added relevant contents in the main text (Page 6).

Secondly, the non-Hermitian behavior, featuring sympathetic resonance resulting from the energy exchange between dual cavities, is the other significant innovation. It contributes to applications such as high-performance sensors and information processing. We have heightened its importance in the main text (Page 9).

Relevant revisions are made as follows:

Page 6 in the main text, “Far-field phonon coupling enables long-distance cavity energy localization and unlocks functionalities in complex systems, such as dual-cavity configurations.”

Page 9 in the main text, “The discovery of energy exchange between dual cavities is not only crucial for the emergence of non-Hermitian phenomena, but also underpins sympathetic resonance at shared frequencies. Such non-Hermitian behavior is vital for high-performance sensing and information processing.”

Reviewer #1: 7. The authors should present numerical methods to calculate the eigenfrequencies of supercells for TESs and TWGMs, e.g., B-A-B elongated and B-inner A-outer hexagonal supercells. Additionally, it should be clarified whether periodic boundary conditions are applied.

Authors: Thank you for this comment on numerical methods. For both TESs and TWGMs, periodic boundary conditions are applied along the x -direction and continuity conditions along the y -direction for the B-A-B supercell, and periodic boundary conditions are also applied to three pairs of parallel sides for the hexagonal supercell. We have added related details in the Methods of main text (Page 11).

Relevant revisions are made as follows:

Page 11 in the main text, “Periodic boundary conditions are applied to the supercell B-A-B along the x -direction, whereas continuity conditions are applied along the y -direction.”

Page 11 in the main text, “Topological whispering gallery modes (TWGMs) are analyzed using a rhombus cavity supercell (side length $7a$) positioned between the inner PnC-B and outer PnC-A (Supplementary Fig. 5a). Periodic boundary conditions are applied to three pairs of parallel sides of the hexagonal supercell, yielding the corresponding eigenfrequencies and displacement

fields (Fig. 2b; Supplementary Figs. 5b and 5c).”

Reviewer #1: 8. How the authors distinguish the transmission peaks or dips in experiment? As we can see the frequency difference between the two peak frequencies in Fig. 3b-d is only about 0.01 MHz versus the central frequency 1.96 MHz.

Authors: Thank you for raising this experimental issue. We were able distinguish transmission peaks or dips with a frequency difference of only 0.01 MHz in the experiment, mainly relying on the following measures:

Firstly, the input signal power was amplified by ~ 37 dB to enable the generation of large ultrasonic pulses and ensure undamaged silicon wafer. At each space point, 512 times of scans was used to reduce the amplitude of white noise.

Secondly, we set the sampling frequency $f_s = 100$ MHz, being over 47 times of target frequency f_c ($1.7 \text{ MHz} < f_c < 2.1 \text{ MHz}$). At each space point, we recorded signal by 10^5 data points, being over 1700 times of target frequency period ($1/f_c$). Therefore, it became possible to analyze wave pulses with fine filtering width towards target frequency f_c , with step variation every 0.001 MHz.

By repeating the above procedures, we can obtain the amplitude of out-of-plane displacement within the target space region. These data were further processed to separate neighboring $|u_z|$ peaks (at cavity edge) or dips (at waveguide outlet). Besides, these procedures also allowed us to get the animation or segment of wave displacement within target space region. We have added relevant experimental methods before Fig. 6 in the main text (Pages 10-11).

Relevant revisions are made as follows:

Pages 10-11 in the main text, “As shown in Fig. 6, the experimental setup comprised a Polytec OFV 2570 laser Doppler vibrometer (LDV), a high-speed camera, a RIGOL DG1032z signal generator, a power amplifier, and a Tektronix DPO4102B-L oscilloscope. Silicon samples were mounted on a supporting plate with a spatial positioning precision of $\sim 5 \mu\text{m}$. A $5 \text{ mm} \times 1 \text{ mm}$ PZT disk was bonded to the back surface of the waveguide inlet using conductive adhesive. A seven-cycle sinusoidal burst signal centered at frequency f_c was generated, amplified, and delivered to the PZT transducer. The number of cycles in the burst was limited to facilitate identification of the incident wave packet as it propagated through the valley metamaterial phononic circuits. The LDV measured the out-of-plane displacement (u_z) within target regions, while the high-speed camera recorded the laser beam position and enabled spatial point selection.

To enhance experimental accuracy, several measures were taken. The input signal power was amplified by ~ 37 dB to generate strong ultrasonic pulses without damaging the silicon wafer. At each spatial point, 512 scans were averaged to reduce white noise. The sampling frequency was set to $f_s = 100$ MHz, more than 47 times the target frequency f_c (1.7–2.1 MHz). At each spatial point, signals were recorded with 10^5 data points, corresponding to more than 1700 periods of $1/f_c$. This allowed fine spectral filtering near f_c with a frequency resolution of ~ 0.001 MHz. By repeating these procedures, we obtained the amplitude of the out-of-plane displacement across the target region, enabling separation of neighboring TWGMs and dips. These procedures also allowed us to reconstruct animations and spatial segments of wave displacement within the region of interest.”

Response to reviewer #2

Reviewer #2: The noteworthy result presented in the manuscript is the demonstration of 'far-field' coupling between a waveguide and cavities in a valley phononic plate crystal, through a 'Dirac strip'. Among the claims, I was not convinced by what the authors call 'synchronized oscillations': in the understanding of this reviewer there is no oscillator in the experiments presented, but resonators excited by a single-frequency driving source (the term 'oscillator' has a clear meaning in physics and engineering and appears to be misused).

The Methods section says that 'direct cavity-to-cavity coupling is prohibited', then why does the main text advertise 'synchronized oscillations in the resonators', whereas clearly the same external source drives both of them?

Authors: Let us first express our thanks for the positive comment on 'far-field coupling'.

We also appreciate the insightful comment on 'synchronized oscillations'. We agree that the term 'synchronized oscillations' does not accurately reflect the response behavior between the two cavities.

As you have stated pointed, in our experiment, the two cavities were not independent 'oscillators', but were resonators simultaneously excited at the same frequency under a driving source. We have therefore replaced the term 'synchronized oscillations' with 'sympathetic resonances' throughout the main text (Pages 1-2, 6-9).

As for the logical contradiction in your proposal between 'direct coupling prohibited' and 'synchronous oscillation', it stems from our misused 'synchronized oscillations'. The 'direct cavity-to-cavity coupling is prohibited' refers to the absence of short-range direct interaction between the two cavities. The observed dual-cavity resonances are mediated by a long-range coupling pathway through the Dirac strips, which enables simultaneous excitation of TWGMs in both cavities by the same external driving source. Therefore, there is no longer a contradiction between 'direct cavity-to-cavity coupling is prohibited' and 'sympathetic resonances'.

Relevant revisions are made as follows:

Page 1 in the main text, “We also combine near- and far-field cavities on the same substrate, and thereby amplify and control non-Hermitian dynamics through loss- and distance-modulated **sympathetic resonances**, directly resolved via **piezo-laser** interferometry.”

Page 2 in the main text, “By co-locating near- and far-field cavities, we observe **sympathetic resonances**—simultaneous resonance in both cavities at a single frequency—and reveal non-Hermitian dynamics modulated by material loss and inter-cavity distance.”

Page 6 in the main text, “This confirms **sympathetic resonances**³³ at each TWGM₃ frequency—i.e., both cavities are resonating simultaneously—validated by the $|u_z|$ field maps (middle/bottom panels in Fig. 4b).”

Page 7 in the main text, “Enhanced coupling and **sympathetic resonances** in the dual-cavity waveguide system.”

Page 8 in the main text, “Space-controlled coupling and **sympathetic resonances** in the dual-cavity waveguide system.”

Page 8 in the main text, “We now modulate inter-cavity coupling by adjusting the horizontal

distance L from $-7a$ to $16a$, probing how **sympathetic resonance** affects transmission dips.”

Page 9 in the main text, “In addition, by integrating dual cavities in both the near and far fields we establish deterministic control over non-Hermitian dynamics through material loss and spatial separation, experimentally observing **sympathetic resonances**.”

Reviewer #2: The discussion of non-Hermitian circuits also sounds off for the same reason: there is dissipative (material, structure) loss and radiation loss, for sure, but there is no gain.

Authors: Thank you for the comment on non-Hermitian system. Truly, there is no gain in our system. We use the concept of "non-Hermitian" for several reasons:

First, the non-Hermitian system is non-conservative, requiring only energy exchange. This can occur with only loss, only gain, or both present simultaneously [Rev. Mod. Phys. 93, 01505 (2021); Nat. Phys. 14, 11 (2018)]. We follow this more general conclusion of "non-Hermitian" system, in the main text and included relevant references (Page 1).

Secondly, although our system lacks gain, it does experience energy loss. Our dual-cavity system not only includes the material loss, but also energy loss to the external environment generated by the dual-cavity system itself, making it intrinsically a non-conservative system.

Thirdly, this work studies the non-Hermitian phenomena based on the energy exchange between two cavities. From this perspective, we derive the theory that considers dual-cavity energy exchange based on the sympathetic resonance. We have added related contents in the main text (Page 9).

Relevant revisions are made as follows:

Page 1 in the main text, “**Non-Hermitian systems are non-conservative and characterized by energy exchange^{15,16}, encompassing three situations: loss only, gain only, as well as the coexistence of both.**”

Page 9 in the main text, “**The discovery of energy exchange between dual cavities is not only crucial for the emergence of non-Hermitian phenomena, but also underpins sympathetic resonance at shared frequencies.**”

Reviewer #2: What the main manuscript calls 'pump-probe' is actually a piezoelectric source and a laser vibrometer, there is no laser pump (while first reading the text I thought laser ultrasound was being used in the experiment but this not the case at all).

Authors: Thank you for pointing out the term 'pump-probe'. The experiments used a piezoelectric excitation source and a laser vibrometer, without involving laser pumping. Our experimental method, more precisely, is a piezo-laser method. To address your concern, we have revised the 'pump probe' to 'piezo-laser' in the main text (Page 1).

Relevant revisions are made as follows:

Page 1 in the main text, “We also combine near- and far-field cavities on the same substrate, and thereby amplify and control non-Hermitian dynamics through loss- and distance-modulated **sympathetic resonances**, directly resolved via **piezo-laser interferometry**.”

Reviewer #2: Strangely, the supplemental information is much clearer and informative than the main manuscript. It is written in simple engineering terms and the figures are clear and not

composed of too many panels. Globally the methodology is consistent, but most of the useful information is given in the SI, such as geometric parameters, band gap frequencies, Dirac cone deformation with angle θ - I personally tend to dislike this way of writing papers, aiming at impact rather than accuracy.

I recommend a reorganization of the manuscript with important figures of the SI included in the main text (SI figures 1, 2, 5 typically). The revision should be made in order to clarify the methodology and to reconsider the claims: there is no reason to oversell what has been achieved, the experiment is rather nice and seems well conducted.

Authors: Thank you for your detailed comments on the content balance between the main text and SI. We agree with you that key information should be prioritized in the main text to ensure accuracy. In response to your specific suggestion, we have systematically reorganized our manuscript.

First, we have combined the original SI Figures 1 and 5 into a new Figure 1 in the main text. At the same time, we have moved the original descriptions of these two figures to the corresponding positions in the main text, and we have removed duplicate sentences. We have taken these contents as the first section of the main text to present the useful information such as sample structures, geometric parameters, and frequency dispersion of the unit cell (Pages 2-3).

Secondly, as the original SI Figure 2 is directly related to the experimental method, we have moved the original SI Figure 2 (now Fig. 6) and related contents (including Tables 1-2) to the Experimental methods section of the main text (Pages 10-11).

In summary, we have reexamined and reorganized the entire manuscript, and improved its completeness and accuracy.

Relevant revisions are made as follows:

Pages 2-3 in the main text, “

Valley metamaterial circuits: three configurations

Figure 1a shows the three valley metamaterial circuits investigated in this work. Sample 1 is a far-field cavity (FFC) waveguide circuit composed of triangular pillars, with side length $s_l = 500 \mu\text{m}$ and height $h_p = 292 \mu\text{m}$ (see Fig. 1b). Samples 2 and 3 correspond to the near-field cavity (NFC) waveguide circuit and dual-cavity circuit, respectively, with $s_l = 526 \mu\text{m}$ and $h_p = 289 \mu\text{m}$. All samples share the same lattice constant $a = 641 \mu\text{m}$ and wafer thickness $e_s = 525 \mu\text{m}$, with geometrical parameters detailed in Table 1 (see Methods). The rhombus cavities have side length $7a$, and the waveguides span $38a$ horizontally. Rightward terminal boundaries are oblique to suppress reflections. The Dirac strips maintain a $3a$ width horizontally, extending vertically 7 layers for FFC structures and 3 layers for NFC structures. In dual-cavity configurations, the NFC and FFC are positioned $7a$ apart horizontally. Simplified models of the three samples are shown in Supplementary Figs. 1 and 2.

We fabricated valley phononic crystal (PnC) plates by etching arrays of triangular pillars in a honeycomb lattice (see Methods). Dirac cones appear at the K point of the Brillouin zone for $\theta = 0^\circ$ (Figs. 1c-d, upper panels), while material anisotropy introduces a minor band gap (insets). Rotating the pillars to $\theta = 20^\circ$ (phase A) and $\theta = -20^\circ$ (phase B) generates broad band gaps for antisymmetric plate waves (Figs. 1c-d, lower panels), with slight frequency shifts of

the Dirac cones between Sample 1 and Samples 2–3.

The unit cell of triangular pillars possesses C_{3v} symmetry when θ equals zero, and topological phase transitions are induced by changing rotation angle θ to break this symmetry. These valley PnCs support the topological states reported previously^{24,29}. We adopt this classic paradigm and fabricate all three samples on silicon wafers to explore novel mechanisms of wave propagation.

Fig. 1 | Samples and dispersion relations for the valley phononic plates. **a** Experimental samples for the FFC-waveguide, NFC-waveguide, and dual-cavity waveguide circuits. **b** PnC plate, unit cell, and Brillouin Zone. Geometrical parameters include lattice constant a , pillar side length s_l , pillar height h_p , silicon wafer thickness e_t , and substrate thickness $e_s = e_t - h_p$. The topological phases are controlled by tuning the rotation angles θ of the triangular pillars. **c, d** Band structures of the unit cell for $\theta = 0^\circ$ (top panels) and $\theta = \pm 20^\circ$ (bottom panels) are shown for Sample 1 in (c) and Samples 2 and 3 in (d). The color scale indicates the ratio of $|u_z|$ to the total displacement $|u_t|$ in the unit cell. Insets show the scheme of unit cells and a zoom-in of the dispersion for $\theta = 0^\circ$.

Table 1. Geometrical parameters of three samples

	Sample 1	Sample 2	Sample 3
	Single far-field cavity	Single near-field cavity	Dual cavity
Lattice constant a (μm)	641		
Wafer thickness e_t (μm)	525		
Pillar height h_p (μm)	292	289	
Pillar side length s_l (μm)	500	526	

Table 2. The recipe for the BOSCH process

Process	Platen power (Watt)	Pressure (mTorr)	SF6 (sccm)	C4F8 (sccm)	Time (s)
Passivation	0	19	0	85	7
Etching	20	37	175	0	12

”

As shown in Fig. 6, the experimental setup comprised a Polytec OFV 2570 laser Doppler vibrometer (LDV), a high-speed camera, a RIGOL DG1032z signal generator, a power amplifier, and a Tektronix DPO4102B-L oscilloscope. Silicon samples were mounted on a supporting plate with a spatial positioning precision of $\sim 5 \mu\text{m}$. A $5 \text{ mm} \times 1 \text{ mm}$ PZT disk was bonded to the back surface of the waveguide inlet using conductive adhesive. A seven-cycle sinusoidal burst signal centered at frequency f_c was generated, amplified, and delivered to the PZT transducer. The number of cycles in the burst was limited to facilitate identification of the incident wave packet as it propagated through the valley metamaterial phononic circuits. The LDV measured the out-of-plane displacement (u_z) within target regions, while the high-speed camera recorded the laser beam position and enabled spatial point selection.

To enhance experimental accuracy, several measures were taken. The input signal power was amplified by $\sim 37 \text{ dB}$ to generate strong ultrasonic pulses without damaging the silicon wafer. At each spatial point, 512 scans were averaged to reduce white noise. The sampling frequency was set to $f_s = 100 \text{ MHz}$, more than 47 times the target frequency f_c (1.7–2.1 MHz). At each spatial point, signals were recorded with 10^5 data points, corresponding to more than 1700 periods of $1/f_c$. This allowed fine spectral filtering near f_c with a frequency resolution of $\sim 0.001 \text{ MHz}$. By repeating these procedures, we obtained the amplitude of the out-of-plane displacement across the target region, enabling separation of neighboring TWGMs and dips. These procedures also allowed us to reconstruct animations and spatial segments of wave displacement within the region of interest.

Fig. 6 | Experimental piezo-laser setup for measuring ultrasonic wave fields in the cavity-waveguide circuit.”

Reviewer #2: As a final note, the experimental results in figures 2 and especially figure 3 are not really successful in demonstrating localized fields established inside the cavities; whatever the experimental reasons it seems the conclusions can only rely on numerical simulations - this should be acknowledged.

Authors: Thank you for your careful review on the experimental results of original Figures 2 and 3 (now Figs. 3 and 4).

First, for FFC-waveguide system, we analyzed the reason why the experimental wave fields in the cavity are not as uniform as the numerical results. This is due to the backscattering caused by the corners of the rhombus cavity, a similar phenomenon is observed in references [Nat. Photon. 17, 386 (2023); Nano Lett. 24, 5570 (2024)]. In addition, there exists boundary effects from the metamaterial to the pure silicon wafer, and the limitations in the number and power of incident pulses. These factors prevent waves from fully propagating along the rhombus path as simulations, making it difficult to clearly capture the local field inside the cavity. Therefore, we acknowledge the full display of the wave field depends on the simulations. Nevertheless, experimental results are still essential to comprehensively demonstrate the performance of the cavity-waveguide system. We have added relevant contents in the main text (Page 6).

Then, similar situation is for experimental wave fields of the dual-cavity waveguide system. In addition to material loss and bulk waves, the experimental characterization of the full wave field along the rhombus path is difficult because of the limited circle and power of the ultrasonic pulses. The ideal display of the local field inside the cavity still relies on simulations, while experimental results are indispensable for further analysis. We have added relevant contents in the main text (Page 7).

Relevant revisions are made as follows:

Page 6 in the main text, “Supplementary Fig. 8 shows TWGM₁ generation at P₁/D₁ perturbed by bulk waves in the Pure PnC, and verifies the experimental perturbation by bulk waves at

P_1/D_1 .”

Page 6 in the main text, “Besides material loss and the influence of bulk waves, the experiment also encounters the effects of backward scattering³² (arising from cavity corners²⁰) and boundary effects at the metamaterial–silicon interface, whereas the input waves are limited in number and pulse power (see Methods). These factors can prevent waves from fully circulating along the ring path as ideally as in simulations. While the full demonstration of wave fields relies on simulations, experiments remain essential to reveal the functional capabilities of the waveguide circuits, serving as a foundation for future applications. Far-field phonon coupling enables long-distance cavity energy localization and unlocks functionalities in complex systems, such as dual-cavity configurations.”

Page 7 in the main text, “Besides material loss, bulk waves are visible beyond the cavities in the wave field of Fig. 4b. As with the FFC circuit, experimental characterization of the full wave field along the ring paths is limited by ultrasonic pulses of finite duration and power. Simulations capture the ideal wave-field behavior, while experiments remain essential to reveal the main factors influencing wave propagation.”

Response to reviewer

Reviewer #1:

In the respond letter, authors point to point response to my comments about the details of this study. After thorough-going analysis and clear description the details of the study, authors reasonably show the significance of this work. In a word, this paper is a careful study and the findings are of considerable interest. This paper is worth to be published.

Author:

Thank you again for your positive comment and recommendation. We are delighted that you recognize our study as a meticulous work with significant findings.

Reviewer #2:

The authors have conducted a very convincing revision. I am highly satisfied with all answers to my comments and suggestions. The correction of terminology, I hope, makes the manuscript clearer regarding the precise achievements. The redistribution of information between main text and supplemental text is now adequate.

Overall this is a very well written manuscript (I did not spot any typo), concise and precise in the methodology. The conclusions are well supported by the results. I believe the manuscript is ready for publication and I recommend acceptance.

Author:

We once again thank you for your careful review. Your professional comments and suggestions have greatly helped us to improve the quality of the present paper.